# Does Improvement in Health-Related Lifestyle Habits Increase Purpose in Life among a Health Literate Cohort?

**DOI:** 10.3390/ijerph17238878

**Published:** 2020-11-29

**Authors:** Shunsuke Kinoshita, Nobutaka Hirooka, Takeru Kusano, Kohei Saito, Hidetomo Nakamoto

**Affiliations:** Department of General Internal Medicine, Saitama Medical University, Saitama 350-0495, Japan; kinoppi@saitama-med.ac.jp (S.K.); t_kusano@saitama-med.ac.jp (T.K.); k_saito@saitama-med.ac.jp (K.S.); nakamo_h@saitama-med.ac.jp (H.N.)

**Keywords:** purpose in life, health-related lifestyle, health promotion, disease prevention, healthy behavior

## Abstract

A growing number of studies have revealed the association between health-related lifestyle habits and purpose in life. However, the mechanism linking the two has not been adequately understood. This study aims to examine the effect of changes in health-related lifestyle habits on purpose in life. A retrospective cohort study was conducted on certified professional specialists of health management. We analyzed the cohort’s demographic information, health-related lifestyle behaviors, reported changes in health-related lifestyle habits (exercise, diet, sleep, and other habits), and purpose in life using a validated tool (Ikigai-9). The cohort was divided into four groups based on the number of reported changes in health-related lifestyles. The purpose in life score was compared among the four groups with and without adjusting for lifestyle. In total, there were 4820 participants. The means (and SD) of the Ikigai-9 score for groups 1, 2, 3, and 4 were 31.4 (6.6), 32.2 (5.6), 32.8 (5.8), and 34.9 (5.4), respectively. There was a statistically significant difference in the Ikigai-9 score among the groups. Healthier changes in lifestyle habits increased perceptions of purpose in life. Both purpose in life and health-related lifestyle habits might be the target factors for disease prevention and health promotion.

## 1. Introduction

Having a purpose in life is fundamental to humans. Several benefits may be gained by enhancing one’s purpose in life [1,2] including the overcoming of physical and psychological disorders, reduction of mortality from multiple diseases, the improvement of health, and increases in life expectancy [3,4,5,6,7]. Japan, one of the most rapidly aging nations in the world, faces the national issue of increased health care costs and the burden of nursing care for the elderly induced by lifestyle diseases [8]. The national health promotion program called Health Promotion Program in the 21st Century (Health Japan 21; HJ21) was implemented to create a society where all nationals can live a healthy and purposeful life [8]. Purpose in life, Ikigai in Japanese, is defined as something to live for, the joy and goal of living [7,9]. It is an important part of the HJ21 goals. Studies have showed that a higher sense of purpose in life is associated with low over-all and cause-specific mortality including cardiovascular disease and coronary heart disease [7,9]. 

Several studies have shown that positive lifestyle choices and behaviors are the foundation of good health [10,11,12]. As a corollary, unhealthy lifestyle habits have been shown to be risk factors for developing noncommunicable diseases (NCDs) [13,14]. Obesity is associated with many different diseases including cardiovascular disease (acute myocardial infarction, angina, stroke), which are major causes of death worldwide [15]. The rise in obesity has brought attention to lifestyle as a prominent cause of disease in modern times [16,17]. Recently, it has also been noted that there is growing evidence that quality and quantity of sleep are associated with NCDs such as cardiovascular diseases [18,19]. In addition, studies have shown that health-related lifestyle interventions are effective in reducing the incidence of many diseases including NCDs such as diabetes, coronary heart disease, and stroke [20,21,22,23,24]. These diseases are also the leading causes of death worldwide. Systematic reviews have shown the positive impact of modification of diet and exercise habits on prevention of NCDs [25,26]. Owing to the high prevalence, incidence, and mortality of NCDs worldwide, disease prevention and health promotion through lifestyle modifications are being addressed in the national health promotion campaigns of many countries [27]. HJ21 specifically emphasizes components of lifestyle such as exercise, diet, sleep, rest, smoking, and alcohol consumption as targets of intervention, since these are known risk factors for major diseases of mortality and comorbidities.

Identifying healthy health-related lifestyle behaviors are not only beneficial for disease prevention but also improve purpose in life [28,29,30,31,32]. Individuals with a greater purpose in life are more likely to self-regulate and engage in a healthy lifestyle. Once they develop a healthy lifestyle, such individuals are likely to avoid lifestyle-related diseases [33]. Typically, the existing literature on purpose in life considers a single health behavior of interest. Hills et al. showed that diet, exercise, and sleep quality are correlated with purpose in life [34]. They also showed a potential mediating and moderating effect of sleep on the purpose. However, the mechanism underlying the association between multiple health-related lifestyle behaviors and purpose in life has not been adequately investigated [1,34,35]. Proposed hypotheses have not yet established or fully explained the causal link between health-related lifestyle habits and purpose in life [35,36]. A clearer understanding of the mechanism, including the causal link will help to prepare effective intervention strategies, for disease prevention and health promotion. 

This study aims to examine the effect of reported changes in health-related lifestyle habits including diet, exercise, and sleep/rest on the purpose in life to better understand the link between health-related lifestyle behaviors and purpose in life. To our knowledge, there is no previous investigation of the effect of degree of changes in multiple health-related lifestyle habits on purpose in life in the context of a national health promotion program such as the HJ21.

## 2. Materials and Methods

### 2.1. Study Design

This is a retrospective cohort study of professional specialists of health management. We conducted a survey on health-related lifestyle behaviors similar to the questionnaire used in the HJ21 [8,37]. All the surveys of lifestyle, changes in health-related lifestyle habits, and purpose in life were administered at the time of study enrollment. The survey was conducted from December 2018 to March 2019. The study description, the survey form, and informed consent form were sent to the specialists of health management and they were requested to participate in the study through mail. Written informed consent was obtained from each participant and the survey forms were answered at participants’ convenience. The completed survey forms were returned through mail. The survey included measuring individuals’ purpose in life, which was done using a validated tool in Japanese, the purposeful life scale (Ikigai-9) [38]. Changes in health-related lifestyle habits in areas of exercise, diet, rest, and sleep habits after becoming certified specialists of health management were also asked through self-administered questionnaires. 

### 2.2. Study Participants

The study population was identified as individuals who are certified specialists in health management obtained from the register of the Japanese Association of Preventive Medicine for Adult Disease (JAPA) [39]. This certification was sponsored by the Ministry of Education, Culture, Sports, Science and Technology, in Japan. The inclusion criterion was certified specialists of health management who actively maintained their knowledge and skills through the continuing professional education provided by the JAPA. We excluded individuals who did not regularly participate in the continuous education provided by the JAPA. Among those individuals who met the inclusion criterion (*n* = 9149), our final sample comprised certified professionals who agreed to participate in the study (*n* = 4820; males = 1630; females = 3190), who were spread out across the country. Written informed consent was obtained from all the participants. These professional specialists of health management are expected to engage with the community and the society they live in and promote health. The ages of the participants ranged from 20 years to 73 years and the average (and SD) was 55.4 (12.2) years. Professional specialists of health management are required to take courses to be eligible for certification [40]. The courses include aspects of health promotion, lifestyle-related diseases, mental health, nutrition, environment and health, physical activity and exercise, emergency medicine and life support, and health care systems. Moreover, to be registered as health specialists, candidates have to pass the final written examination conducted by the JAPA. Additionally, the JAPA encourages specialists to participate in numerous activities so that they are able to conduct health promotion workshops and lectures and organize health-related activities.

### 2.3. Variables and Measurements

The variables in the study were demographic data, lifestyle behaviors, changes in health-related lifestyle habits, and purpose in life. These were measured using a formerly validated tool (Ikigai-9). Participants were asked about changes in their eating, exercising, and rest/sleep habits after they became certified professionals of health management. The survey questions about the lifestyle behaviors and changes in health-related lifestyle habits were the same as those used in the HJ21. These questionnaires were based on one of the oldest national health surveys around the world, the National Health and Nutrition Survey (NHNS) conducted by the Japanese Government [41]. The NHNS serves as a major national database for risk factors for noncommunicable diseases in Japan [42]. The sampling frame used for these national surveys (from which we obtained comparison data for the data obtained in the present study) was a list of all residential census enumeration areas in Japan, stratified into all 47 prefectures in the country; the sampling method was stratified random selections at the prefecture, census area, and household levels, respectively [43]. The response rates for the lifestyle survey element of the NHNS (performed annually) ranged from 59% to 68% between 2003 and 2013. Lifestyle behaviors regarding eating/dieting, exercise/physical activity, sleep, rest, smoking, and alcohol intake were asked about in the questionnaire. The method of the self-administered questionnaires used in the NHNS was studied to test the validation and reliability for a food frequency questionnaire, including questions about alcohol consumption [44,45], anthropometry [46], and physical activity [47]. There were 11 health-related lifestyle questions, among which four were dichotomous scale (“Intention to maintain ideal weight,” “Exercise,” “Manage lifestyle to prevent disease,” “Smoking”). For these items, a score of “1” was assigned for an unhealthy lifestyle behavior and a score of “4” was assigned for a healthy lifestyle behavior. Such a mechanism was followed to ensure impartiality. Questions regarding alcohol consumption included the type of alcohol, amount of drinking, and frequency of drinking. This information allowed for if participants drink more than the recommended amount (less than 20 gm per day in average) used in the HJ21. Then, a score “1” was assigned if participants drink more than a recommended level and “4” was assigned if they drink less than 20 gm per day on average. The remaining six health-related habits (“Reading nutritional information labels,” “Maintaining a balanced diet in daily life,” “Intention for exercise,” “Stress,” “Rest,” and “Sleep”) were answered using a 4-point Likert type scale. Thus, “4” (most favorable) to “1” (least favorable) were assigned for these variables. Scores for each question were added to obtain a composite score of health-related lifestyle of each participant.

For reported changes in health-related lifestyle habits, they were asked to rate their responses on a 5-point Likert-type scale, where 1 indicated significantly improved, 2 indicated somewhat improved, 3 indicated no change, 4 indicated somewhat worse, and 5 indicated significantly worse. Subsequently, lifestyle changes were grouped based on improvements (accounted by “improved significantly” and “improved somewhat”) in the three categories (eating, exercise, and rest/sleep). Groups 1, 2, 3, and 4 indicated reported improvements in no category, 1 category, 2 categories, and 3 categories based on the number of improvements, respectively. 

Meanwhile, Ikigai-9 consists of nine questions on the various purposes in life including emotions towards one’s life, attitudes towards one’s future, and acknowledgement of one’s existence and each question was rated on a 5-point Likert scale, where 1 indicated strongly disagree and 5 indicated strongly agree. The total score of Ikigai was calculated by adding all the nine scores. Age, weight, height, BMI, and purpose in life scores were recorded numerically.

### 2.4. Analysis

Descriptive statistics (mean, average, SD, range) were used to describe the study participants’ characteristics. Ikigai-9 scores as the outcome variable were compared among the four groups in terms of the number of changes in health-related lifestyle habits as the explanatory variable, both by adjusting no variables (with the ANOVA test), and by adjusting the age and the lifestyle variables (with the ANCOVA test). Post-hoc tests by controlling type 1 errors were performed using the Games-Howell test. We performed a multiple linear regression test to investigate if there was any moderating effect of lifestyle behaviors on the association between reported changes in lifestyle habits and purpose in life. All statistical tests were two-tailed and *p* < 0.05 was considered as being statistically significant. IBM SPSS Statistics (Version 26.0. Armonk, NY, USA) was used for the analysis. 

## 3. Results

The total number of study participants enrolled was 4820. Table 1 shows the characteristics of the study participants’ lifestyle behaviors and reported changes in health-related lifestyle habits. There were 898, 646, 945, and 2331 study participants in groups 1, 2, 3, and 4, respectively. There were statistically significant differences of proportions of eating, exercise, and rest/sleep among the 4 groups. The means (SD) of Ikigai-9 score for groups 1, 2, 3, and 4 were, 31.4 (6.6), 32.2 (5.6), 32.8 (5.8), and 34.9 (5.4), respectively (Table 1).

We found a statistically significant difference among the groups (*F*(3, 4816) = 103.3, *p* < 0.001). When controlling for the familywise error rate (Type 1 error) and considering unequal sample size among groups, pairwise post-hoc tests (the Games-Howell test) indicated a statistical significance between groups 1 and 3, groups 1 and 4, groups 2 and 4, and groups 3 and 4 (Table 2). The statistically significant difference in the scores of purpose in life among the groups remained after controlling for the age and lifestyle scores, (*F*(3, 4815) = 26.6, *p* < 0.001).

Multiple linear regression was performed to investigate the moderating effect of lifestyle behaviors on purpose in life showed statistically non-significant interaction between changes in health habits and health-related lifestyle after adjusting for ages. The regression analysis also indicated that behavioral changes in lifestyle habits and baseline lifestyle behaviors significantly explained the changes in purpose in life (Table 3).

## 4. Discussion

The main finding of this study was that individuals who reportedly improved their health-related lifestyle habits obtained a higher score on purpose in life. The findings remain after adjusting for covariates. This is the first study to show that individuals who improved health-related lifestyle habits maintain a strong sense of purpose in life in the context of the national health promotion. The results support the concept of the HJ21 that aims to increase quality of life through lifestyle modification [8]. 

At present, two major theories have been proposed to explain the mechanistic link between purpose in life and health behaviors. In addition, a recent discussion included the role of many other factors such as societal milieu, psychological factors, humanity, life events, etc. in the causal mechanism [48]. Elucidating the influence of these variables on lifestyle and purpose in life ultimately is hoped to illustrate the better understanding of the complex mechanism. One of the main causal mechanisms responsible for the relationship between purpose in life and health behaviors is proposed that purpose in life is the promoting factor for healthier lifestyle [35,36]. Purpose in life was shown to be associated with health-related lifestyle in several studies [3,28,29,30,31,32]. There are many studies to show the benefit of lifestyle modification. Some showed the beneficial effect of dietary and exercise intervention on NCDs [25,26]. Abbate et al. showed in their systematic review that dietary intervention combined with exercise had the most beneficial effect on cardiovascular events as compared with diet or exercise intervention alone. Sleep and rest are also considered important target to combat NCDs [18,19]. Hill et al. reported multiple components of the healthy lifestyle as mediators of the association between the purpose in life and health [34]. Thus, it is hypothesized that purpose in life promotes healthier lifestyle behaviors including diet, exercise, sleep/rest, and the combinations, which ultimately prevents diseases and improves health.

However, this is in contrast to the hypothesized theory that purpose in life positively influences health-related lifestyle behaviors [35,36], which results in the prevention of lifestyle-related disease and promotes health. In the meantime, a few studies demonstrated that engaging in healthier lifestyles behavior predicts greater purpose in life [48,49]. Thus, reverse causality [35], purpose in life as an outcome rather than a predictor, is another proposed mechanism responsible for the relationship between purpose in life and health-related lifestyle behaviors. Although the data were retrospective, our findings revealed that individuals who improved health-related lifestyle habits gained a higher purpose in life. There is little evidence to support the reverse causality, but our data supports the notion of reverse causality. This study also shows a linear relationship between the degree of change in health-related lifestyle habits and purpose in life. Individuals who changed their lifestyle habits in one (group 2), two (group 3), or three (group 4) categories of exercise, diet, rest and sleep, statistically significantly gained 0.76, 1.43, and 3.53 higher scores on purpose in life, compared to the scores of no change (group 1), respectively. The results also showed that the linear relationship between changes in the lifestyle habits and purpose in life remained after considering the moderator effect of the baseline lifestyle behaviors. The linear relationship or dose-response between the predictors and outcome variables is one of the criteria in establishing the causal relationship [50,51], and our findings support the causal link that positive changes in health-related lifestyle habits increase the purpose in life. 

This does not discard the main causal relationship between purpose in life and lifestyle behaviors. There is growing support in literature on the main causal mechanism of purpose in life, on health outcomes through lifestyle and behaviors [35,36]. Considering both existing findings and the results of the present study, a mechanistic link between health-related lifestyle behaviors and purpose in life can be considered to be bidirectional. Therefore, it seems important to consider the interventions to both, lifestyle and purpose in life, as targets for health promotion.

While the causal mechanisms between health-related lifestyle behavior and purpose in life need further investigation, our study highlights on the strategy for health promotion program. It is critical to reduce the burden of health care costs and nursing care of elderly in a rapidly aging society [52]. Previous studies have shown that if the goal of the HJ21 regarding longer healthy longevity is achieved through the program, then the proportion of disability payments will decrease by 1% per year and ultimately 2500–5300 billion Japanese yen (equivalent to 24–50 billion U.S. dollars) will be saved in long-term care and medical care costs [53]. In the meantime, it is of value to improve quality of life for citizens [54,55]. The study results support the potential benefit of intervention to health-related lifestyle behaviors in the context of a national health promotion program, since people who changed their health-related lifestyle habits positively maintained a strong sense of purpose in life. Thus, our study results support the strategy of the health promotion at the national level such as the HJ21, which emphasizes the primary prevention of lifestyle-related chronic diseases and enhances active health to increase quality of life through lifestyle modification.

Limitations of the current study include reporter bias in several measures due to self-reporting [56]. Future studies may corroborate the measurement in the lifestyle behaviors by using robust approaches and/or using objective devices such as widely available wearable devices to determine if the changes measured with the objective record are significant in increasing purpose in life [57,58]. In addition, the design was a retrospective study and there remain unknown and some potential confounding factors such as socioeconomic variables, education level, income, or marital status, in the analysis. However, major influencing factors such as age and lifestyle behaviors, were adjusted for the analysis. Thus, the causal relation between the health-related lifestyle behaviors and purpose in life should be further investigated prospectively, incorporating these factors into measurements and analysis. Finally, the external validity of the study may not be high, since participants in the study were highly educated and held expertise in the field of health management. The study population was restricted to those who are health literate. Future studies should target wider populations in investigating the mechanism of impact of changes in health-related lifestyle habits on purpose in life.

The study’s strength included a large sample size that allowed meaningful analysis to investigate the associations between changes in health-related lifestyle habits and purpose in life. The cohort study design, being retrospective, helped to better understand this causal relationship. 

## 5. Conclusions

Healthier changes in health-related lifestyle habits increase one’s purpose in life. The results support the potential benefit of lifestyle modification. With the known effect of lifestyle changes on preventing many diseases, and goals of the national health promotion, a higher sense of purpose in life, is achievable through lifestyle changes. The national health promotion program promotes a sustainable and vital society where all nationals live healthier and fulfilling lives. 

## Figures and Tables

**Table 1 ijerph-17-08878-t001:** Changes in health-related lifestyle habits in the groups of participants.

	Total (*n* = 4820)	Group 1 (*n* = 898)	Group 2 (*n* = 646)	Group 3 (*n* = 945)	Group 4 (*n* = 2331)
Sex (%)					
Male	33.8	31.7	29.6	33.1	36.1
Female	66.2	68.3	70.4	66.9	63.9
Age ** (Ave years, SD)	55.4 (12.2)	52.2 (12.2)	53.1 (11.3)	53.9 (11.9)	57.9 (12.0)
BMI (Ave kg/m^2^, SD)	21.9 (3.3)	21.7 (3.3)	21.9 (3.4)	21.8 (3.6)	22.0 (3.2)
Eating habit ** (%)					
Significantly improved	15.4	0	4.5	9.3	26.9
Somewhat improved	59.7	0	63.9	81.1	72.8
No change	24.6	98.9	31.3	9.6	0.3
Somewhat worse	0.2	0.7	0.2	0.1	0
Significantly worse	0.1	0.4	0.2	0	0
Exercise habit ** (%)					
Significantly improved	12.5	0	0.9	5.7	23.3
Somewhat improved	51.3	0.1	16.1	61.8	76.4
No change	35.7	98.6	81.7	32.3	0.2
Somewhat worsened	0.3	0.9	1.1	0.2	0
Significantly worsened	0.1	0.4	0.2	0	0
Rest/sleep ** (%)					
Significantly improved	9.7	0	0	2.1	19.2
Somewhat improved	48.3	0	16.4	36.7	80.5
No change	40.9	97.7	81.6	59.5	0.2
Somewhat worsened	0.9	1.9	1.9	1.5	0
Significantly worsened	0.1	0.4	0.2	0.2	0
Lifestyle score ** ([95% CI])	35.5 [35.4–35.6]	33.7 [33.4–34.0]	33.9 [33.5–34.2]	34.8 [34.6–35.1]	36.9 [36.7–37.0]
Ikigai-9 score ** ([95% CI])	33.5 [33.3–33.7]	31.4 [31.0–31.8]	32.2 [31.7–32.6]	32.8 [32.5–33.2]	34.9 [34.7–35.2]

Groups 1, 2, 3, and 4 indicate groups of participants who improved in zero, one, two, and three areas of change in lifestyle after registration, respectively. Ave; average, CI; confidence interval, and SD; standard deviation. ** indicates *p* < 0.01.

**Table 2 ijerph-17-08878-t002:** Pairwise post hoc comparisons of purpose in life score among the groups.

	Group 1	Group 2	Group 3
Group 2	2.37	-	-
Group 3	4.96 *	2.30	-
Group 4	15.52 **	11.35 **	9.83 **

Groups 1, 2, 3, and 4 indicate groups of participants who improved in zero, one, two, and three areas of change in lifestyle after registration, respectively. t-statistics are shown in the table. * indicates *p* < 0.05 and **; *p* < 0.01.

**Table 3 ijerph-17-08878-t003:** Evaluating associations between reported changes in lifestyle habits and purpose in life in multiple linear regression.

	B	SE B	*β*
Constant	10.79	1.67	
Changes in lifestyle habits	1.38	0.57	0.27 *
Lifestyle behavior score	0.58	0.05	0.39 *
Interaction term ^a^	−0.02	0.02	−0.17
Age	0.01	0.01	0.02

^a^ Interaction term between the variable of reported changes in lifestyle habits and health-related lifestyle. B, SE, and β indicate unstandardized coefficient, standard error of unstandardized coefficient, and standardized coefficient, respectively. * indicates *p* < 0.05.

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
