# Peer review of "Does Improvement in Health-Related Lifestyle Habits Increase Purpose in Life among a Health Literate Cohort?"

_ijerph, 2020, doi:10.3390/ijerph17238878_

Round 1

Reviewer 1 Report

The authors have largely changed the manuscript in line with the comments. After revision, the manuscript quality is average. It has some new value and can be published. I leave the final decision to the editor on whether to accept the manuscript for publication.

Reviewer 2 Report

Dear author(s)

Thank you for considering carefully my previous suggestions, comments and observations.

I believe the overall quality of your manuscript was improved a lot. Therefore, at this stage, I would recommend only minor spell checks.

All the best

This manuscript is a resubmission of an earlier submission. The following is a list of the peer review reports and author responses from that submission.

Round 1

Reviewer 1 Report

The manuscript addresses an important issue, namely the relationship between the strong purpose in life and general concern for one`s own well-being. As it can be guessed, there is a correlation between these variables, which the authors showed on a large group of respondents. The purpose and method of conducting the study leave many questions, therefore the following list of questions must be answered in the redrafted text before publishing the manuscript.

  1. How long before the survey did the respondents start to care of their well-being? Is the observed increase in the purpose in life the fresh result of recent changes or it is an established trend?
  2. Were there any other good/optimistic or bad/pessimistic circumstances in the life of the respondents that could influence the purpose in life?
  3. What experience took place earlier in the respondents` life: the current level of the purpose of life, or the current level of well-being? The answer to this question should ultimately determine whether a change in the purpose of life causes a change in habits or vice versa.

Author Response

Reviewer 1 comments

Comments and Suggestions for Authors

The manuscript addresses an important issue, namely the relationship between the strong purpose in life and general concern for one`s own well-being. As it can be guessed, there is a correlation between these variables, which the authors showed on a large group of respondents. The purpose and method of conducting the study leave many questions, therefore the following list of questions must be answered in the redrafted text before publishing the manuscript.

(Response) Thank you for reviewing our manuscript and providing valuable comments. We tried to answer all the comments below and revised the manuscript based on the comment and replies.

How long before the survey did the respondents start to care of their well-being? Is the observed increase in the purpose in life the fresh result of recent changes or it is an established trend?

(Response)

The respondents were certified specialists of health management. The certification process required studies of 6 core modules and submission of reports on every module. Passing a written test was also required to be certified. The process usually takes a year. Then, each specialist performed activities of health management in the society they belong to at their own pace. The period of time the respondents that took care of their well-being was thus considered to be from the time they studied the modules to the point that they answered the questionnaire in the study. The score of purpose in life in the study (outcome) was measured also at the point of study participation (i.e., answering the questionnaire). We added details on the measurements of time regarding the purpose in life in the method section (lines 77-79).

Were there any other good/optimistic or bad/pessimistic circumstances in the life of the respondents that could influence the purpose in life?

(Response)

There might be systemic influences on the purpose in life caused by factors such as earthquakes, COVID-19, etc. These may have affected some of the respondents and influenced their association between the purpose in life and lifestyle changes. While we did not ask or include each of these factors, we adjusted the lifestyle variable, which included perceived stress at the time of answering the questionnaire. We included the discussion of systemic influence by other factors including psychological and societal factors that may affect the association between changes in lifestyle habits and purpose in life in the discussion section using references (lines 189-191).

What experience took place earlier in the respondents` life: the current level of the purpose of life, or the current level of well-being? The answer to this question should ultimately determine whether a change in the purpose of life causes a change in habits or vice versa.

(Response)

Thank you for the thoughtful comments. As pointed out, experiences that took place earlier in the study participants’ life play an important role as confounding or direct modifier when analyzing the relation between the changes in lifestyle habits and purpose in life. Thus, in the discussion, we added life experience as a potential factor influencing the causal link between purpose in life and lifestyle habits.  

Reviewer 2 Report

Dear author(s)

Thank you for your work. I went carefully through your paper and I found your study very interesting. Nevertheless I believe you can further improve it. Hopefully my suggestions will be helpful.

Best wishes

Introduction

  • Across (lines 30-33), please describe briefly what is intended by purpose in life. Even a description in broad terms may be of interest to the readers.
  • I fear that the word “is” is a misprint/typo (line 41).

Materials and methods

Study design

  • Could you please explain better how many questionnaires were proposed to participants in the study? It is not clear if you administered only one survey investigating all aspects at once or different surveys investigating different aspects at different times.
  • In particular, if the second case holds, could you please illustrate in more detail the process of questionnaire administration?
  • Why did you choose certified professional specialists of health management for your survey?
  • How did you select the participants? What was for example your sampling strategy?
  • How did you administer the survey? What format (paper, electronic,…)? Where and under what conditions did they fill in the survey, if available?
  • Moreover, it is not clear when the respondents became certified professional specialists, and I think this is a quite relevant aspect as in practical terms I assume you consider it as a part of the repondents’ respective purpose in life. Time may somehow alter their perception in this sense and this may have a crucial impact on the main findings of your study. Maybe it could be helpful for you that you try to include a figure (e. g. a time line) in this section of the methods to help the readers follow the process.

Study participants

  • Please, better rephrase the sentence reported here (lines 66-68).
  • In the second part of the sentence in (line 75), please consider the possibility to revise the language a bit.

Variables and measurements

  • As you describe many types of different scales, could you please provide some examples of your questions by investigated dimension or, even better, could you please add an appendix with the full questionnaire(s)?

Discussion:

Could you please try to illustrate some examples of potential practical implications that your findings may have for health management and policy?

Author Response

Introduction

Across (lines 30-33), please describe briefly what is intended by purpose in life. Even a description in broad terms may be of interest to the readers.

(Response)

Thank you for the comments. In the study, purpose in life was used in the context of the Japanese National Health Promotion in the 21st Century (HJ21). Thus, Ikigai as concept was used in this study. Thus, we addressed the definition in the introduction.

I fear that the word “is” is a misprint/typo (line 41).

(Response)

We appreciate that you pointed out this typo. We have corrected this sentence.

Materials and methods

Study design

Could you please explain better how many questionnaires were proposed to participants in the study? It is not clear if you administered only one survey investigating all aspects at once or different surveys investigating different aspects at different times.

In particular, if the second case holds, could you please illustrate in more detail the process of questionnaire administration?

(Response)

We thank you for this comment. There may be a lack of clarity owing to the present form of the description of the questionnaire. However, all aspects in the study were asked about at the same time. There were no surveys at different times in the study. Thus, we edited to make sure this was made clear in the revised manuscript of the design section.

Why did you choose certified professional specialists of health management for your survey?

How did you select the participants? What was for example your sampling strategy?

How did you administer the survey? What format (paper, electronic,…)? Where and under what conditions did they fill in the survey, if available?

Moreover, it is not clear when the respondents became certified professional specialists, and I think this is a quite relevant aspect as in practical terms I assume you consider it as a part of the respondents’ respective purpose in life. Time may somehow alter their perception in this sense and this may have a crucial impact on the main findings of your study. Maybe it could be helpful for you that you try to include a figure (e. g. a time line) in this section of the methods to help the readers follow the process.

(Response)

Thank you for your inquiry on the administration of the questionnaire. The entire survey process including the timing of certification and survey administration were added to the method section.

Study participants

Please, better rephrase the sentence reported here (lines 66-68).

(Response)

We clarified the study participants in this sentence.

In the second part of the sentence in (line 75), please consider the possibility to revise the language a bit.

(Response)

Thank you for the comments. We have changed the sentence to ensure clarity.

Variables and measurements

As you describe many types of different scales, could you please provide some examples of your questions by investigated dimension or, even better, could you please add an appendix with the full questionnaire(s)?

(Response)

All the questions for lifestyle, changes in health-related lifestyle behaviors, and Ikigai (purpose in life) were presented in the method section with the Likert scale. To make them clearer, we submit the questionnaire as a supplement.

Discussion:

Could you please try to illustrate some examples of potential practical implications that your findings may have for health management and policy?

(Response)

We have discussed the practical implication for the policy, the Japanese National Health Promotion in the 21st century, whose context was the main focus in this study. We have also expanded on the discussion regarding health management program in general.

Reviewer 3 Report

Thank you very much for considering me as a reviewer of the article entitled "Does Improvement in health-Related lifestyle habits increase purpose in life among a healht literate cohort?". After reading it carefully I have some suggestions to make:

In general, the study needs a great effort to deepen the conceptual and methodological aspects that will allow us to have an adequate basis on which to develop the results and discussion of the study. It is significant that the authors have not reviewed the journal's publication rules regarding format and citation.

1. Introduction:

In my opinion, the authors make too brief and superficial an introduction without directly addressing the main concepts of the study such as lifestyles or purpose in life. For example, in the first line, they mention the existence of multiple benefits in improving purpose in life. However, they do not mention them until the end of the first paragraph. It is advisable to synthesize this paragraph by referring to the benefits directly, since the second and third sentences are repetitive. In addition, the authors mention the existence of different benefits, but when quoting the benefits they do not include quotes 1 and 2.

How is the purpose in life defined or would the authors define it?

With regard to lifestyles, there are multiple definitions within the literature, and this is a very much discussed topic. How do the authors define it? It is advisable to go deeper into the definition and characteristics of "health-related lifestyles". What types of interventions have been carried out, have they all been equally effective, and is there any previous evidence applied in the field of health management? Many questions arise when reading the introduction.  Many ideas are expressed that should be supported by the previous evidence as in lines 41-43.

The aim of the study does not correspond to what was developed in the study, it should be better defined. How is the change in lifestyles assessed? Are measurements made at two different times?

2. Materials and methods

2.1. Study design

How valid is the questionnaire used to evaluate lifestyles? The authors note that the Ikigai-9 questionnaire is validated. however, they do not mention anything about lifestyles. How many items are composed in each study?

They mention the lifestyle categories in the summary but do not develop them or mention anything in the introduction about which lifestyle is more decisive than others.

2.2. Study participation

The text contains some symbols underlined in yellow. Do they mean anything?

Has any sampling been done to select the participants? Have other socio-demographic measures been evaluated such as marital status, educational level (even if a minimum qualification is stressed), do they have any diseases or are they currently physically active? These aspects may be relevant to analyse life changes. For example, a single and a married person do not have the same purpose and lifestyles.

2.3. Variables and measurements

This section is difficult to understand, they comment on lifestyles, then talk about the purpose of life and then return to lifestyles. It is advisable to structure the study variables appropriately.

Do the two questionnaires use the same type of Likert scale and how reliable are they?  It could be evaluated using Cronbach's Alpha to get an estimate.

One aspect that is not clear is whether they identify four different lifestyles (exercise, diet, rest and sleep), and group rest and sleep into the same group. How do they identify four groups to segment participants, if there are three as indicated by the authors?

They then mention that there are eleven health-related lifestyles including alcohol or tobacco intake. Why are they not mentioned above? In which categories are these aspects included? Because they do not relate to the categories mentioned above, it is a different one.

What other nominal or ordinal variables have been assessed?

2.4. Analysis

What is the level of significance that was established?

Why do the authors not conduct a cluster analysis to determine the groups based on their responses in all groups and not according to the number of lifestyles that have changed? Already the changes in the categories are not grouped together, i.e. they are not made on the basis of the specific lifestyle in which they are changing. There may be two subjects in the same established category who have had changes in different lifestyles.

3. Results

Table 1 is outside the established margins. In addition, it is recommended to include the percentages, since for many readers knowing the proportion helps to better understand the results than N. It would be advisable to make a description of each group. In the title of the column of each group it would be advisable to indicate the number of participants in each group.

They indicate only one variability of the ANOVA test, which variable is it? They should all be specified in the table according to the variable as well as the significance. It is not determined in which variables there are differences and between which groups. This would help the reader to better observe the analyses and the differences that are not clearly established by the authors.

Table 2 duplicates the information, using one side of the diagonal only. The results are not properly interpreted. Which differences are more significant? Between which groups?

4.Discussion

The aim of the study is not specified. As mentioned above, it is not very clear what the authors' purpose is in this study. They mention the theory that was previously stated but that has not been clearly reflected in the introduction. The study does not contain enough theoretical background to be able to support a hypothesis.

The discussion is limited to reiterate what is mentioned in the introduction, they do not justify adequately their results by checking them enough with the previous findings because they do not provide enough information about them.

What can be done in future studies to correct the limitations that the authors have stated?

5. Conclusions

The conclusions are very brief, and the novelty or relevance of the study is not apparent. What is new about this study compared to what already exists?

What are the practical applications of the study?

6. References

There are spaces between words in different references (Lines 196, 197, 211, for example).

Author Response

Comments and Suggestions for Authors

Thank you very much for considering me as a reviewer of the article entitled "Does Improvement in health-Related lifestyle habits increase purpose in life among a health literate cohort?". After reading it carefully I have some suggestions to make:

In general, the study needs a great effort to deepen the conceptual and methodological aspects that will allow us to have an adequate basis on which to develop the results and discussion of the study. It is significant that the authors have not reviewed the journal's publication rules regarding format and citation.

  1. Introduction:

In my opinion, the authors make too brief and superficial an introduction without directly addressing the main concepts of the study such as lifestyles or purpose in life. For example, in the first line, they mention the existence of multiple benefits in improving purpose in life. However, they do not mention them until the end of the first paragraph. It is advisable to synthesize this paragraph by referring to the benefits directly, since the second and third sentences are repetitive. In addition, the authors mention the existence of different benefits, but when quoting the benefits they do not include quotes 1 and 2.

(Response)

Thank you for your comments suggesting how we can make the introduction more in depth with descriptions of the purpose in life and lifestyle behaviors. We have now provided more evidence to support the benefit of the purpose in life and lifestyle changes in the context of disease prevention, which is the main concept and goal of the study.

How is the purpose in life defined or would the authors define it?

(Response)

Thank you for this important question. There may be differences in the definition of the purpose in life in different studies. Since our study was performed in the context of HJ21, we used the definition widely accepted in Japan. We have included the definition of the purpose in life in the introduction.

With regard to lifestyles, there are multiple definitions within the literature, and this is a very much discussed topic. How do the authors define it? It is advisable to go deeper into the definition and characteristics of "health-related lifestyles". What types of interventions have been carried out, have they all been equally effective, and is there any previous evidence applied in the field of health management? Many questions arise when reading the introduction.  Many ideas are expressed that should be supported by the previous evidence as in lines 41-43.

(Response)

Since the main topic of the study is part of a national health promotion, HJ 21, to prevent NCDs and increase QOL, part of which is the purpose in life, we have incorporated more detail about the influence of lifestyle and purpose in life on NCDs in the introduction and  discussion, supported by previously reported evidence (lines 34-44 & 181-185).

The aim of the study does not correspond to what was developed in the study, it should be better defined. How is the change in lifestyles assessed? Are measurements made at two different times?

(Response)

Lifestyle habit changes were determined by the respondents by looking back on the period before being certified and at the time of the survey (lines 110-112). They were measured once at the time of the survey in the study (lines 75-78).

  1. Materials and methods

2.1. Study design

How valid is the questionnaire used to evaluate lifestyles? The authors note that the Ikigai-9 questionnaire is validated. however, they do not mention anything about lifestyles. How many items are composed in each study?

(Response)

Our study was performed under the concept and context of HJ21. We used the same questionnaire that was used in the National Health and Nutrition Survey, which is the basis of the HJ21’s goal. In the method section, we added information about the questionnaire, which has been used in the National Health and Nutrition Survey (lines 112-115).

They mention the lifestyle categories in the summary but do not develop them or mention anything in the introduction about which lifestyle is more decisive than others.

(Response)

The lifestyle habits investigated in the study were those related to noncommunicable chronic diseases. This is the focus of HJ21 and thus, the focus of our study, too. We added detail on health-related lifestyle habits such as exercise, diet, rest and sleep in the introduction.

2.2. Study participation

The text contains some symbols underlined in yellow. Do they mean anything?

(Response)

These do not mean anything and have been removed.

Has any sampling been done to select the participants? Have other socio-demographic measures been evaluated such as marital status, educational level (even if a minimum qualification is stressed), do they have any diseases or are they currently physically active? These aspects may be relevant to analyze life changes. For example, a single and a married person do not have the same purpose and lifestyles.

(Response)

It is quite important to measure socio-economic demography. However, in the current study we did not include marital status or educational level. Further, we have discussed the unmeasured factors including socioeconomic status as limitations of the study and suggested that they be considered in the future studies.

2.3. Variables and measurements

This section is difficult to understand, they comment on lifestyles, then talk about the purpose of life and then return to lifestyles. It is advisable to structure the study variables appropriately.

(Response)

We have revised the structure of the study variables as suggested by the reviewer (lines 109-138).

Do the two questionnaires use the same type of Likert scale and how reliable are they?  It could be evaluated using Cronbach's Alpha to get an estimate.

(Response)

We only administered one questionnaire at one point. The questionnaire used the Likert scale as described in the variables section. There were two types of scales for lifestyle. One was a yes-no scale including questions such as “Do you smoke?”. The second had 5 ratings: 1-significantly improved; 2- somewhat improved; 3- no change; 4- somewhat worse; and, 5- significantly worse. We have clarified this by describing the method of administering the questionnaire in the study design (2.1 Study Design) and scale in the variables and measurements section (2.2 Variables and Measurements).

One aspect that is not clear is whether they identify four different lifestyles (exercise, diet, rest and sleep), and group rest and sleep into the same group. How do they identify four groups to segment participants, if there are three as indicated by the authors?

(Response)

In the category of rest and sleep, the questionnaire asked about these two lifestyle habits at the same time. However, this does not to appropriately measure each of the habits as stated. Nonetheless, this specific question was used in the HJ21 format. Since we followed the HJ21 guide, we used the question as it was set in the HJ21. Therefore, these two habits were grouped as one variable.

They then mention that there are eleven health-related lifestyles including alcohol or tobacco intake. Why are they not mentioned above? In which categories are these aspects included? Because they do not relate to the categories mentioned above, it is a different one.

(Response)

The health-related lifestyles being measured in the 11 questions were used as covariates. The explanatory variable, lifestyle habit changes (not the lifestyle themselves), was derived from the hypothesis tested in the study to investigate the influence of behavioral changes on the purpose in life. As previously mentioned as a response to a reviewer’s comment, we revised the description of the explanatory variable, covariates, and outcome variable in the variables and measurements section (lines 117-138). This includes how we assign the score for each lifestyle questions including alcohol drink.

What other nominal or ordinal variables have been assessed?

(Response)

We did not measure nominal or ordinal variables other than the ones mentioned in the variables and measurements section. All of the nominal and ordinal variables were explained. Thus, the sentence was deleted.

2.4. Analysis

What is the level of significance that was established?

(Response)

In the method section, we added the level of significance, which was p < .05.

Why do the authors not conduct a cluster analysis to determine the groups based on their responses in all groups and not according to the number of lifestyles that have changed? Already the changes in the categories are not grouped together, i.e. they are not made on the basis of the specific lifestyle in which they are changing. There may be two subjects in the same established category who have had changes in different lifestyles.

(Response)

One of the important steps to investigate causal relation is to verify the dose response between the explanatory factor and outcome, as seen in Hill’s criteria. Thus, we wanted to show that the more changes in the health-related lifestyle, the higher purpose in life is for this reason. Further, it is important to show how each lifestyle change influences the purpose in life. However, it is also important to show aggregated/clustered effect (that may have a synergistic effect) of behavioral changes on the purpose in life. Therefore, we categorized the group based on the number of lifestyle changes, not lifestyle themselves, and used them as the explanatory variables. We have explained this analytic approach in a more precise manner in the analysis section (lines 140-144). 

  1. Results

Table 1 is outside the established margins. In addition, it is recommended to include the percentages, since for many readers knowing the proportion helps to better understand the results than N. It would be advisable to make a description of each group. In the title of the column of each group it would be advisable to indicate the number of participants in each group.

(Response)

We have revised the table according to the suggestions. Now Table 1 includes the description and proportions of each variable.

They indicate only one variability of the ANOVA test, which variable is it? They should all be specified in the table according to the variable as well as the significance. It is not determined in which variables there are differences and between which groups. This would help the reader to better observe the analyses and the differences that are not clearly established by the authors.

(Response)

The ANOVA test was conducted for the sake of comparing the purpose in life scores (an outcome variable) among the groups of changes in lifestyle habits categorized by the number of lifestyle changes (0 to 3). The ANOVA test and the following post-hoc tests both with and without correction of type 1 error were shown in the manuscript. This may have caused confusion so we have only showed the type 1 error controlled comparison between each pair of the two groups. Then, the analysis section was changed to describe the post-hoc tests with type 1 error using the Games-Howell tests.

Table 2 duplicates the information, using one side of the diagonal only. The results are not properly interpreted. Which differences are more significant? Between which groups?

(Response)

Table 2 showed the post-hoc test of the ANOVA as titled by showing each t-statistics of the analysis of the purpose in life in the groups with significance (* p < 0.05 and ** p < 0,01). We tried to illustrate that there were significant differences between each of the comparisons and indicated which differences were relatively small or large by showing the t-statistics among each pair of the comparisons. Since there was a repetition of information in the table and the information may cause confusion as the reviewer pointed out, we have changed the table to only show one side of the diagonal only for type 1 error-controlled values.

4.Discussion

The aim of the study is not specified. As mentioned above, it is not very clear what the authors' purpose is in this study. They mention the theory that was previously stated but that has not been clearly reflected in the introduction. The study does not contain enough theoretical background to be able to support a hypothesis.

(Response)

As commented in the analysis section, we have included a clearer statement describing the specific aim of the study with more detail introductory discussion in the introduction section (lines 67-70).

The discussion is limited to reiterate what is mentioned in the introduction, they do not justify adequately their results by checking them enough with the previous findings because they do not provide enough information about them.

(Response)

Thank you for pointing out the inadequacy of the discussion in terms of justifying our analysis based on the results. We have incorporated more evidence to support our discussion to reach the conclusion.

What can be done in future studies to correct the limitations that the authors have stated?

(Response)

Thank you for this query. We have described possible approaches to correct the limitations in the discussion section (lines 219-220 & 223-226).

  1. Conclusions

The conclusions are very brief, and the novelty or relevance of the study is not apparent. What is new about this study compared to what already exists?

(Response)

We have added information on the novelty and relevance of the study based on the discussion in the conclusions (lines 208-217).

What are the practical applications of the study?

(Response)

We included a discussion on the practical implications and potential applications in the discussion section. Then, the conclusion was revised based on the new discussion.

  1. References

There are spaces between words in different references (Lines 196, 197, 211, for example).

(Response)

Thank you for this comment. We deleted the spaces in the references and also did an overall revision of the section.

Round 2

Reviewer 3 Report

I appreciate the effort and appreciate the changes made to the document, however, I consider the amendments to be insufficient. Here are my suggestions:

In general, the document has serious deficiencies in the structure of the information, which prevents it from being presented in a clear and concise manner. Moreover, the rules of the journal, as in the previous version, are not complied with, with numerous defects.

1. Introduction

The introduction presents duplicate information, reiterated at different points. For example, Lines 31 -32 with lines 33-34. The information could be better structured so that the introduction has a clear line of argument.

It is important to establish what is considered lifestyle, since there are multiple conceptions in the literature. A lifestyle should not be defined based on a series of specific habits since there are many others that could be included in the conception. As indicated in line 40, these are indicative factors, but not all the factors that exist. This statement should be referenced on the basis of previous studies.

The purpose in life is defined, but this part should be better structured. But, only obesity is paid attention to as NCDs? Are other ncds not included? In line 49 appears "[new ref]".

Why is obesity receiving greater attention at present? It will have to be derived from some fact that needs to be established. Are there any studies that have not had lifestyle benefits after an intervention?

Statements such as those made on lines 57-59 should be made on the basis of previous studies supporting them.

2. Material and methods

2.1 Study Design

How is the value of the index of purpose in life calculated? It should be more specific for those who do not know the Ikigai-9 method.

2.2 Study Participants

The population referred to at the beginning of the section is 9149 individuals or is it only a group covered by the study?

It would be more appropriate to establish inclusion/exclusion criteria that specify the aspects to be considered in order to be eligible, since they are distributed in different points of the section.

2.3 Variables and Measurements

It would be advisable to divide the explanation of each section into paragraphs to make it clearer. They indicate that the instruments were validated. However, the reliability of these instruments is not specified through any index established for this purpose.

What is the criteria established to assign the scores or determine the degree of alcohol consumption? Especially the latter is important since there are different estimates of consumption according to the country.

The criterion used to categorise the groups according to the change in lifestyles is not considered adequate and objective enough. As these were self-administered questionnaires, it was not possible to determine for the participant the difference between improving something or improving it significantly, the answer to the participant's opinion is fully exposed, there could be differences between two specialists.

For example, it is not the same to indicate that my physical activity has improved considerably because instead of two days a week I exercise four days. Even if it is twice as many days and for the subject it may be significant, this increase does not necessarily mean that I am complying with the recommendations for weekly physical exercise as set out by the WHO.

Moreover, in its relationship with NCDs such as obesity it would not imply that it has really significant physical benefits on the subject that would reduce the risks of factors or diseases associated to obesity. This type of modification should have been determined with some accepted scale as IPAQ could establish the amount of physical activity and the level of it that could be done before or after being a more objective criteria to determine if the change has been significant or not.

2.4 Analysis

Why are only the quantitative variables compared and not the qualitative ones? The existence of differences in lifestyles and not only in purpose in life should also be compared.

The statistical analyses raised are not directly related to the objective and hypothesis raised in the study. An ANOVA test will observe differences between groups but does not imply that there is a causal relationship between the variables or concepts being addressed. To establish causal relationships, other types of statistics such as regression or SEM should be applied.

3. Results

The results are not properly interpreted, treating only the data related to the purpose in life. The proportion of participants belonging to each group is not described or detailed. In addition, the presentation of Table 1 shows different shortcomings.

Why are frequency and percentage presented in gender, but only percentage in lifestyles? It is the same type of variable. Or why do you mark "-" on some variables when there is no data and "0" on others?

Statistically significant differences are not marked in the Tables.

Why was a self-adjustment of the covariates performed?

4. Discussion

The discussion does not specify the objective of the study to introduce the main findings. As mentioned above it is not appropriate to discuss the results presented with the cause relationship between lifestyle changes and a higher score on the purpose with life. The appropriate statistics are not made.

As I have reasoned before the deficiency in the determination of lifestyle change such as exercise cannot imply compliance with the minimum recommendations for positive health effects. Therefore, the study hypothesis put forward cannot be accepted.

The discussion needs to go into greater depth on the results obtained. For example, why are lifestyles assessed and data presented when their results are not commented on or discussed?

The practical applications are more related to the conclusions of the study than to the discussion.

5. Conclusions

The relationship between the study objective and the analyses is not sufficiently clear to be able to determine the true contribution of this study. The value of the study is not provided, it just repeats what has been indicated in results and discussion.